# FaceComposer: A Unified Model for Versatile Facial Content Creation

**Jiayu Wang**[*1], **Kang Zhao**[*1], **Yifeng Ma**[*2], **Shiwei Zhang**[1], **Yingya Zhang**[1],
**Yujun Shen**[3], **Deli Zhao**[1], **Jingren Zhou**[1]

[1]Alibaba Group    [2]Tsinghua University    [3]Ant Group

{wangjiayu.wjy,zhaokang.zk,mayifeng.myf,zhangjin.zsw,yingya.zyy,jingren.zhou}@alibaba-inc.com

{shenyujun0302,zhaodeli}@gmail.com

## Abstract

This work presents `FaceComposer`, a unified generative model that accomplishes a variety of facial content creation tasks, including text-conditioned face synthesis, text-guided face editing, face animation *etc.* Based on the latent diffusion framework, `FaceComposer` follows the paradigm of compositional generation and employs diverse face-specific conditions, *e.g.*, Identity Feature and Projected Normalized Coordinate Code, to release the model creativity at all possible. To support text control and animation, we clean up some existing face image datasets and collect around 500 hours of talking-face videos, forming a high-quality large-scale multi-modal face database. A temporal self-attention module is incorporated into the U-Net structure, which allows learning the denoising process on the mixture of images and videos. Extensive experiments suggest that our approach not only achieves comparable or even better performance than state-of-the-arts on each single task, but also facilitates some combined tasks with one-time forward, demonstrating its potential in serving as a foundation generative model in face domain. We further develop an interface such that users can enjoy our one-step service to create, edit, and animate their own characters. Code, dataset, model, and interface will be made publicly available.

## 1 Introduction

Due to the rapid development of generative models, such as diffusion models (DMs) [10, 15, 47], VAEs [21], GANs [6] and flow models [5] in the computer vision area, automatic content creation has recently received an increasing amount of attention for its real-world applications. Benefiting from these generative models, facial content creation as a critical part has recently achieved impressive progress and simultaneously shows great application potential, *e.g.*, virtual digital human, artistic creation, and intelligent customer service.

Existing face generative models [20, 23, 39, 61] are usually developed as highly customized systems, meaning that one model can only handle one task. However, this design poses two significant challenges: 1) hard to accomplish complex tasks, such as integrating face creating, editing and then animating the generated face in a single step; 2) redundant consumption of memory and computation. For example, one needs to train and save a number of models to build a multi-functional system, and perform complicated inference processes. The challenges could inevitably limit its further applications and development.

To tackle these problems, we propose compositional `FaceComposer` in this work, a unified model that is capable of simultaneously tackling versatile facial tasks, including face generation, face editing,

---

[*]Equal contribution.

face animation, and their combinations. Specifically, we decompose a given face into multi-level representative factors, *e.g.*, Identity Features, Projected Normalized Coordinate Code (PNCC) [64] and Text2Face (T2F) Embeddings, and then train a powerful latent diffusion model conditioned on them to compose the input face. This design provides the model with exceptional controllability over facial content creation, enabling a seamless process from generating a face to utilizing it. In particular, we have additionally incorporated temporal self-attention to enable joint learning from both images and videos simultaneously. To optimize `FaceComposer`, we finally gather a high-quality large-scale multi-modal face dataset, including 1.1 million face images from pre-existing datasets and 500 hours of meticulously cleaned talking face videos.

Extensive quantitative and qualitative results demonstrate that `FaceComposer` achieves exceptional performance in various facial content creation tasks. `FaceComposer` surpasses previous state-of-the-art methods in face generation, face editing, and audio-driven face animation in terms of the most widely-used evaluation metrics. Furthermore, we have devised intricate yet imaginative tasks to showcase the advantages of our diverse condition composition.

## 2   Related work

**Face generation.** The goal of face generation [34, 53, 56, 57] is to generate photo-realistic face images. Among them, StyleGAN series [17–19] boost the generation quality by introducing controllable and interpretable latent space. After that, some variants [51, 63, 42, 41, 52] are proposed for further quality improvement. One of them, TediGAN [51], maps text into the StyleGAN latent space to make the text-guided face generation. Another example is LAFITE [63], which presents a language-free training framework with a conditional StyleGAN generator. Recently, diffusion models [10, 32, 33, 38, 37] become more and more popular in image synthesis area due to their strong generative abilities. We find fine-tuning a pre-trained diffusion model will bring comparable or even superior performance to the GAN paradigm.

**Face editing.** Face editing [20, 23, 25, 27, 54] aims to manipulate the face images guided by text descriptions or masks. Similar to face generation, StyleGAN-based methods [51, 29, 49, 65] show remarkable performance in face editing. Besides TediGAN, StyleClip [29] combines representation ability of CLIP [31] and generative power of StyleGAN to optimize the editing direction. In contrast to the above methods, CollDiff [13] trains the diffusion model with multi-modal conditions (text + mask), in order to make them complementary. It is noted that TediGAN and CollDiff both support face generation and editing, which differ from ours in the following aspects: 1) TediGAN is essentially designed as a bespoke system, i.e., it can only complete one task (generation or editing) in one model inference. 2) CollDiff uses multiple diffusion models (one model for one condition), leading to low efficiency in training and inference stage. 3) `FaceComposer` only use one diffusion model in a unified framework, and can finish different facial creation combinations in one model inference.

**Face animation.** Face animation [39, 36, 50, 55, 35, 30, 61, 62, 26, 44] intends to make the target face move according to a driving video or audio. Video-driven methods [39, 36, 50, 55, 35] focus on modeling the motion relationships between source and target face. For example, FOMM [39] decouples appearance and motion and introduces keypoints representation to support complex motions. PIRenderer [36] employs 3DMM to better control the face motions. Compared with keypoints and 3DMMs, the PNCC we used contains more dense and intuitive facial structure information, making it easier to be learned by the model. Audio-driven methods [30, 61, 62, 26, 44] pay more attention to the lip synchronization (lip-sync) between the audio and the mouth of target face. MakeItTalk [62] disentangles the content and speaker information to control lip motions. By learning from a powerful lip-sync discriminator, Wav2lip [30] improves lip-sync accuracy. PC-AVS [61] generates pose-controllable talking faces by modularizing audio-visual representations. However, all of them only consider intra-frame information for reconstruction, ignoring inter-frame relationship, which is fixed by adding a temporal attention module in our `FaceComposer`.

## 3   FaceComposer

Recent days have witnessed the powerful generative ability of Latent Diffusion Models (LDMs) [37], which is consequently equipped as the backbone of our `FaceComposer`. We model various facial content creations as a multiple-condition-driven denoising process. Fig. 1 shows the overview of

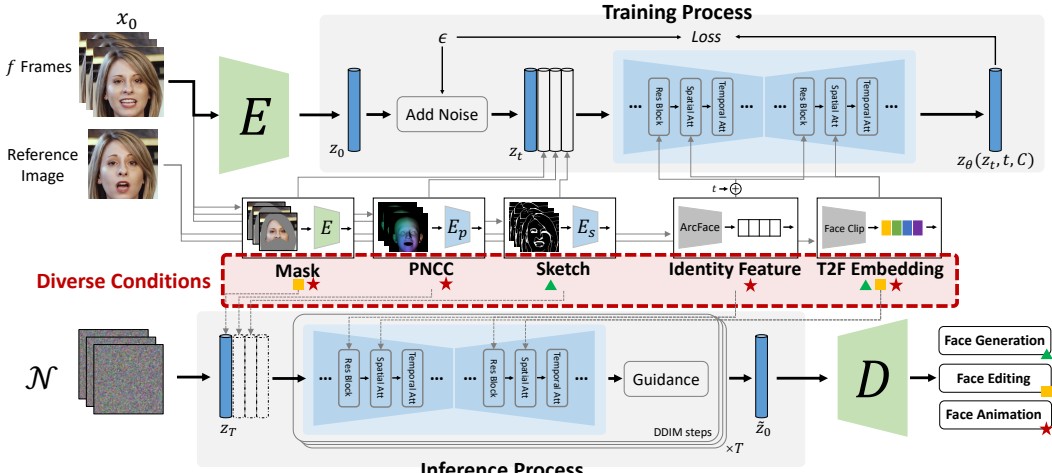

Figure 1: The framework of `FaceComposer`, which takes $f$ frames and five face-related conditions as input, uses LDMs to predict the noise added in the latent space. We can combine diverse conditions to finish face generation/editing/animation or their combinations. For example, the green △ conditions are for face generation, yellow □ for face editing, and red ☆ for face animation.

`FaceComposer`. In the unified framework, the inputs of different tasks are expressed as diverse conditions, including Mask, PNCC, Sketch, Identity Feature and T2F Embedding. We achieve versatile facial content creations through condition combinations (Sec. 3.2). Different from the standard LDMs that is only designed for image generation, `FaceComposer` supports both static and dynamic content creations, *i.e.* our dataset contains both images and videos. So we introduce a temporary attention module into LDMs for the two modalities joint training (Sec. 3.3 and Sec. 3.4).

## 3.1 Preliminaries

We denote the input $f$ frames as $\boldsymbol{x}_0 \in \mathbb{R}^{f \times 3 \times H \times W}$, where $H$ and $W$ are height and width of input frames (we set $H = W = 256$ in experiments).

**Latent diffusion model.** To save the computational resources of DMs, we follow LDMs to encode frames into latent space: $\boldsymbol{z}_0 = E(\boldsymbol{x}_0) \in \mathbb{R}^{f \times C_{latent} \times h \times w}$ with a pre-trained image encoder $E$, where $C_{latent}$ means the dimension of latent space, $h$ and $w$ are set to 32 in practice. And in the end of denoised process, the final $\widetilde{\boldsymbol{z}}_0$ will be mapped into pixel space with image decoder $D$ : $\widetilde{\boldsymbol{x}}_0 = D(\widetilde{\boldsymbol{z}}_0) \in \mathbb{R}^{f \times 3 \times H \times W}$. In the latent space, the diffusion model $\boldsymbol{z}_\theta$ can be parameterized to predict the added noise:

$$\mathcal{L}_{simple} = \mathbb{E}_{\boldsymbol{z}, \boldsymbol{\epsilon}, t, \boldsymbol{C}} \left[ \| \boldsymbol{\epsilon} - \boldsymbol{z}_\theta(a_t \boldsymbol{z}_0 + \sigma_t \boldsymbol{\epsilon}, \boldsymbol{C}) \|_2^2 \right], \tag{1}$$

where $\boldsymbol{C}$ denotes the condition, $t \in \{1, ..., T\}$, $\boldsymbol{\epsilon} \in \mathcal{N}(0, 1)$ is the random Gaussian noise, $a_t$ and $\sigma_t$ are two scalars related to $t$. We freeze $E$ and $D$, and start from a pre-trained LDMs.

**Compositional generation pipeline.** As a pioneering work of compositional generation, Composer [12] decomposes an image into eight conditions to improve the controllability of image synthesis, inspiring us to treat the inputs of different facial content creations as multiple conditions, i.e., $\boldsymbol{C}$ in Eq.1 is a condition set. And we adopt the same *guidance directions* as [12]: $\hat{\boldsymbol{z}}_\theta(\boldsymbol{z}_t, \boldsymbol{C}) = \omega \boldsymbol{z}_\theta(\boldsymbol{z}_t, \boldsymbol{c}_2) + (1 - \omega) \boldsymbol{z}_\theta(\boldsymbol{z}_t, \boldsymbol{c}_1)$, where $\omega$ is the guidance weight, $\boldsymbol{c}_1$ and $\boldsymbol{c}_2$ are two subsets of $\boldsymbol{C}$, respectively.

## 3.2 Diverse face-specific conditions

**Condition decomposition** We convert different inputs into the following five face-specific conditions.

*Mask:* Mask is used to force `FaceComposer` to generate or edit a face in a certain region. Based on the nine parsing areas of face [22], we randomly mask one or all of them. When $f > 1$, different frames mask the same region. Before taken as a condition, masks are also mapped into latent space through $E$.

Table 1: Versatile creations based on condition compositions. *M*, *S*, *PNCCs*, *ID* and *T2F* are short for Mask, Sketch, PNCC sequence, Identity Feature, and T2F Embedding, respectively.

| Single Creation | | Versatile Creations | |
|---|---|---|---|
| Task | Conditions | Task | Conditions |
| face generation | ① *T2F* 
 ② *S* 
 ③ ... | face generation+animation | ① *PNCCs+T2F* 
 ② *PNCCs+ID* 
 ③ ... |
| face editing | ① *M+T2F* 
 ② *M+S* 
 ③ ... | face generation+editing | ① *ID+M* 
 ② *ID+T2F* 
 ③ ... |
| face animation | ① *M+T2F+PNCCs* 
 ② ... | face generation+editing+animation | ① *ID+T2F+PNCCs* 
 ② ... |

*PNCC:* PNCC [64] represents the geometric information of the face, a pre-defined PNCC sequence can effectively guide facial animation generation. We use FLAME Fitting [24] to extract the PNCC of all frames. Different from the mask, the distribution of PNCC differs significantly from the original frame, so we add a trainable module $E_p$ to encode the PNCC.

*Sketch:* Sketch describes the contours of different parts of the face (e.g., face shape, eye size, mouth location), it contains local details with low semantics. We adopt the same method as [12] to extract the sketches. Similar to PNCC, we input sketch information into a trainable $E_s$ module to obtain the condition.

*Identity Feature:* Identity Feature indicates the identity attribute, excluding trivial information (such as hair color, texture, expression). It can direct the model to generate a face with the specified ID. We use ArcFace [3] to get the Identity Feature.

*T2F Embedding:* Text2Face (T2F) Embedding has two functions: 1) complementing Identity Feature with detailed information; 2) enabling text control. Specifically, during the training stage, T2F Embedding is extracted from reference image with Face Clip [60] to feed facial details into `FaceComposer`. In the denoising process, besides the reference image, we can also obtain T2F Embedding from text prompt with an extra prior model (similar to DALL-E 2 [33]).

**Conditioning mechanisms** Considering Mask, PNCC and Sketch represent the spatial local information of frames, they are all extracted from $x_0$ and concatenated with $z_t$ in the channel dimension. In contrast, Identity Feature and T2F Embedding define the global semantic information, hence we get them from the reference image, and add the projected Identity Feature into time embedding, serve T2F Embedding as *key* and *value* for cross attention module of $z_\theta$ (corresponding to Spatial Att in Fig. 1). Note that, except for PNCC, which needs to be fitted in advance, all left conditions are extracted on-the-fly. We adopt a similar condition training strategy to Composer: setting 0.5 dropout probability for each condition, 0.1 to drop all conditions, and 0.1 to reserve them all.

**Condition composition** As we mentioned above, we can support versatile tasks: face generation, face editing, face animation and their combinations by combining our diverse conditions.

We list some representative creations in Tab. 1. Taking face animation as an example, we use the PNCC sequence (predicted by audio), Mask (masking the mouth region of the target face), and T2F Embedding (providing texture information from reference image) to generate Talking Head. It is worth mentioning that `FaceComposer` can finish face generation + editing + animation with a one-time forward by conditioning on Identity Feature (guiding face generation), T2F Embedding (extracted from editing prompt) and PNCC sequence (obtained from reference video). More details are shown in Sec. 4.3.

## 3.3 Temporal self-attention module

In order to create static and dynamic contents simultaneously, we prepare a multi-modal database, consisting of both images and videos. And we argue that joint image-video training is important

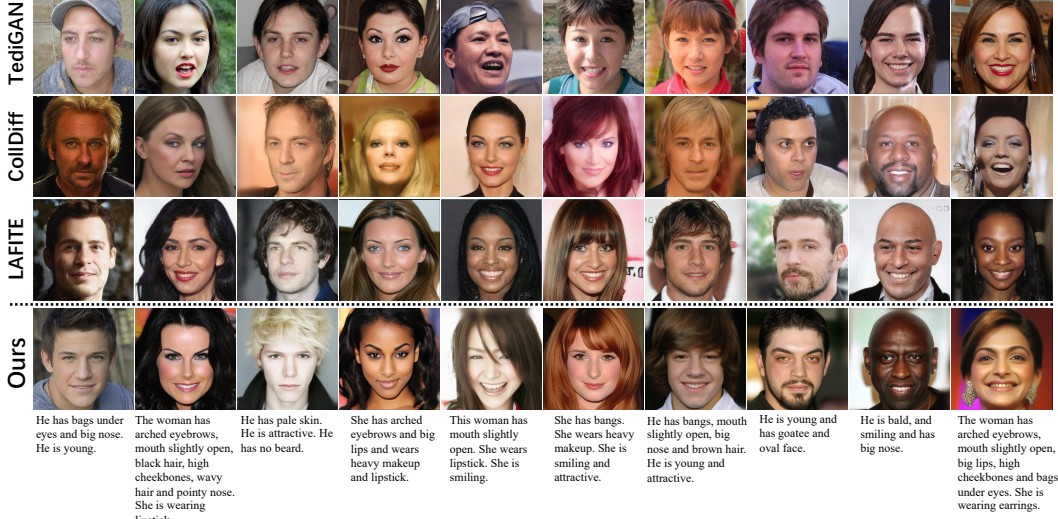

Figure 2: The qualitative results of face generation.

for our `FaceComposer`, considering face image will align facial content and text description, while video will link spatial and temporal information of face sequences.

Inspired by [40, 11], we introduce a temporary self-attention (TSA) layer into LDMs (corresponding to Temporal Att in Fig. 1), and select half batch samples from images (i.e. $f = 1$ for $0.5B$) and the other half from videos (i.e. $f = 5$ for left $0.5B$) to make the joint strategy fully benefit each other within each batch (see ablation study in Sec. 4.4), where $B$ is the total batch size. Assuming the input of TSA is in the shape of $B \times f \times C_{inter} \times h' \times w'$, it will be arranged to $\{B \times h' \times w'\} \times f \times C_{inter}$ before entered into TSA, where $C_{inter}, h', w'$ are the intermediate channel dimension and feature map size. When $f = 1$, TSA degrades to an identity transformation.

### 3.4 Multi-modal face database

To empower `FaceComposer` with image and video generation capabilities, we construct a high-quality large-scale multi-modal face database comprising 1.1 million face images with text annotations and approximately 500 hours-long talking face videos.

**Image data.** To construct the image part of our database, we carefully clean up LAION-Face [60] and merge the cleaned dataset with CelebA-HQ [16] and FFHQ [17]. We clean up LAION-Face using two approaches. Firstly, We use CLIP [31] to filter out the image-text pairs whose text descriptions do not match the images. Specifically, for each image-text pair, we compute the cosine similarity between CLIP features extracted from the image and the text and filter out the pair if the similarity is lower than a predefined threshold. Secondly, we use an off-the-shelf face detector [4] to detect faces in images and filter out images with no faces detected. Finally, we obtain the cleaned LAION-Face dataset. It contains 1 million face images with corresponding text captions.

**Video data.** To construct the video part of our database, we collect talking face videos from Youtube, BBC television, and some other web data. We manually clean collected videos to ensure high video quality and audio-visual coherence. Our collected talking face dataset includes more than 500 hours 720P~1080P videos with the audio track. We will release the dataset when the paper is made public. For more details, please refer to *Supplementary Material*.

## 4 Experiments

### 4.1 Experimental setup

**Implementation details.** During the training, our model starts from a pre-trained LDMs*, and is further trained on our multi-modal face database through a joint training mechanism. To enable text

---

*https://github.com/Stability-AI/stablediffusion

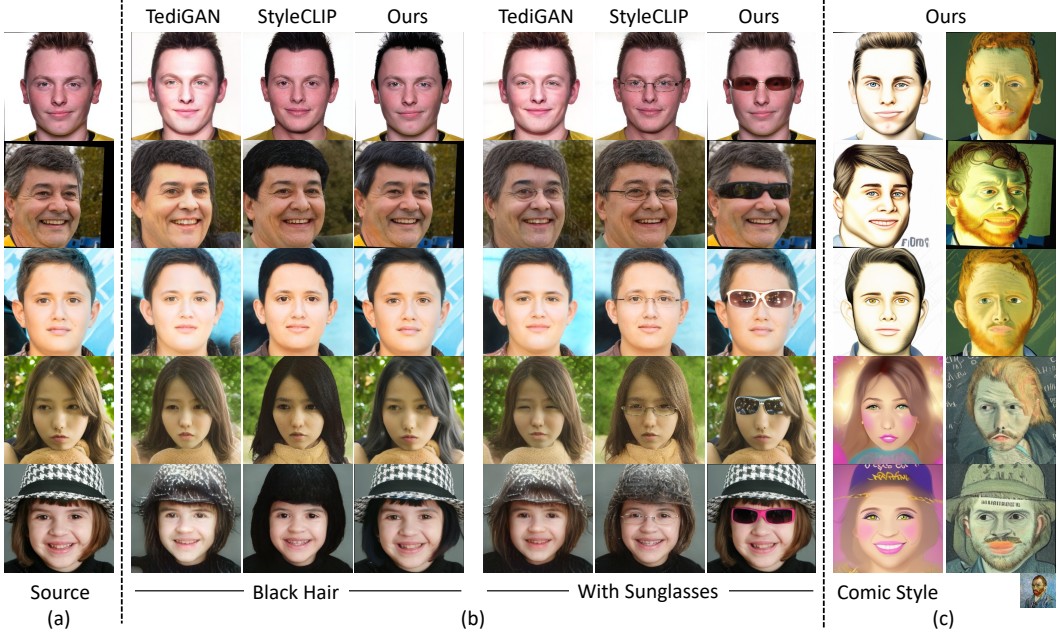

|  | TediGAN | StyleCLIP | Ours | TediGAN | StyleCLIP | Ours | Ours |
| Source (a) | — Black Hair — (b) | | | — With Sunglasses — | | | Comic Style (c) |

Figure 3: The qualitative results of face editing. (a) The source images. (b) The images whose attributes are edited through text descriptions by different methods. (c) The stylized results generated by `FaceComposer`. `FaceComposer` is able to edit the style of images using text (the left column) or a style reference image (the right column).

control, we trained a $1B$ parameter prior model for projecting captions to T2F Embeddings. For the LDMs, we pretrain it with $1M$ steps on the full multi-modal dataset using only T2F Embeddings as the condition, and then finetune the model for 200K steps with all conditions enabled. The prior model is trained for $1M$ steps on the image dataset.

**Evaluation tasks.** We evaluate `FaceComposer` on face generation, face animation and face editing tasks, which respectively using the Multi-Modal CelebA-HQ [51], HDTF [59] + MEAD-Neutral (a subset of MEAD [45] that only contains the neutral facial expression videos), and the randomly selected images from both CelebA and Non-CelebA datasets with randomly chosen descriptions.

**Evaluation metrics.** For face generation task, we adopt Fréchet inception distance (**FID**) [9] to measure the image quality. Since FID cannot reflect whether the generated images are well conditioned on the given captions, we choose to adopt **R-precision** [53] as another evaluation metric. For face editing task, we compute the identity similarities (**IDS**) between the input faces and the edited faces to measure the identity consistency. For face animation task, we adopt the Landmark Distance around mouths (**M-LMD**) [1], the Landmark Distance on the whole face (**F-LMD**), the structural similarity (**SSIM**) [48], the Cumulative Probability of Blur Detection (**CPBD**) [28], and the confidence scores of SyncNet [2] (**Sync$_{conf}$**) as evaluation metrics. To facilitate a more equitable comparison between methods generating only the mouth region (visual dubbing methods) and those generating the entire face (one shot methods), we computed the SSIM and CPBD for the mouth region, denoted as **SSIM-M** and **CPBD-M**, respectively. For more details about the experimental settings, please refer to *Supplementary Material*.

## 4.2 Comparisons on single task

In this subsection, we demonstrate the performance of `FaceComposer` on different tasks, including static content creation, *e.g.*, face generation and editing, and dynamic content creation task, *e.g.*, face animation. Note that for the face animation task, we use audio to drive face motions. We train an Audio2PNCC model to generate PNCCs with lip motions controlled by input audio.

**Face generation.** We compare our method with TediGAN [51], LAFITE [63] and CollDiff [13]. We generate $30,000$ images for each method using randomly selected unseen text descriptions. Tab. 2 shows that our method achieves the best results in terms of both FID and R-precision. Such results

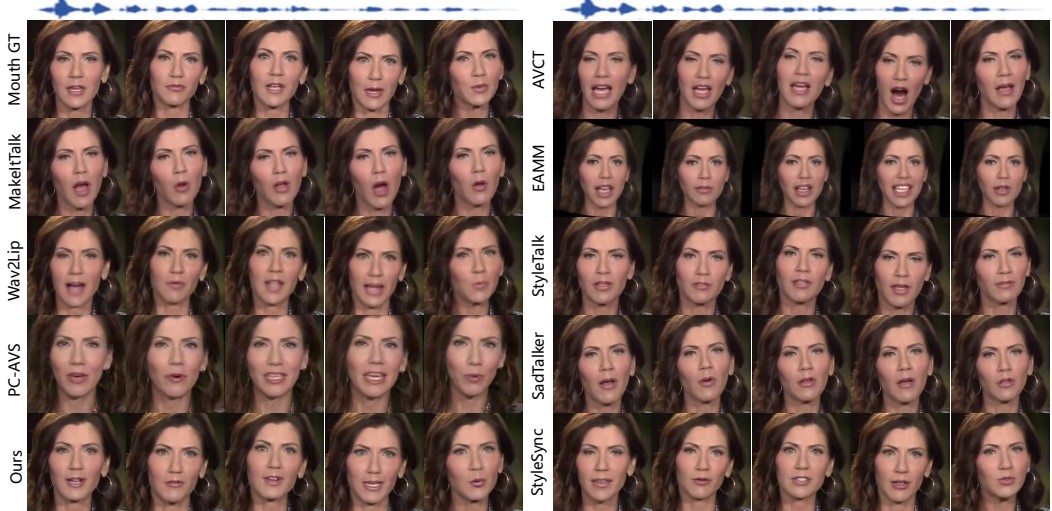

Figure 4: Qualitative results of face animation. It can be seen that `FaceComposer` not only achieves accurate lip-sync but also produces high-fidelity results in the mouth area.

Table 2: Results of face generation.

| Method | FID↓ | R-precision(%)↑ | Accuracy↑ | Realism↑ |
|---|---|---|---|---|
| TediGAN [51] | 107.14 | 44.96 | 2.80 | 3.55 |
| CollDiff [13] | 98.76 | 67.41 | 3.43 | 2.96 |
| LAFITE [63] | 12.54 | 81.94 | 4.07 | 3.88 |
| Ours | **11.34** | **86.63** | **4.86** | **4.67** |

Table 3: Results of face editing.

| Method | IDS↑ | Accuracy↑ | Realism↑ |
|---|---|---|---|
| TediGAN [51] | 0.62 | 2.92 | 2.09 |
| StyleClip [29] | 0.75 | 4.27 | 2.87 |
| Ours | **0.94** | **4.58** | **4.47** |

demonstrate the effectiveness of massive training data. The qualitative comparison is shown in Fig. 2. We can observe that other methods can generate text-relevant results. However, their results lack some attributes contained in the input captions. On the other hand, our results achieve higher fidelity while maintaining text-visual coherence.

We also evaluate the **Accuracy** and **Realism** through a user study, in which the users are asked to judge which image is more realistic, and more coherent with the given captions. We randomly sampling 30 images with the same text conditions and collect more than 20 surveys from different people with various backgrounds. Compared with the state-of-the-arts, `FaceComposer` achieves better accuracy and realism values as shown in Tab. 2, which proves that our methods can generate images with the highest quality and text-relevance.

**Face editing.** To evaluate the face editing performance of `FaceComposer`, we compare our method with TediGAN [51] and StyleClip [29]. Results are shown in Tab. 3, and we can see that our method achieves better results in all metrics (accuracy and realism come from user study, following the same scheme in Tab. 2), which proves that our methods can generate images with the highest quality and text-relevance. We show some visual results in Fig. 3. As we can see, our methods can perform face editing more precisely, and the achieved results are the most photorealistic and most coherent with the given texts.

**Face animation.** We compare our method with state-of-the-art methods including MakeItTalk [62], Wav2Lip [30], PC-AVS [61], AVCT [46], EAMM [14], StyleTalk [26], SadTalker [58], and StyleSync [7]. The samples of the compared methods are generated either with their released codes or with the help of their authors. As shown in Tab. 4, `FaceComposer` achieves the best performance in most metrics. The strong performance in SSIM, SSIM-M, CPBD, and CPBD-M metrics indicates the high quality of videos generated by `FaceComposer`. Given that Wav2Lip employs SyncNet as a discriminator during its training phase, it reasonably attains the highest confidence score from SyncNet, surpassing even the ground truth. Numerous prior arts [7, 58, 26, 8, 43] have reported that Wav2Lip, despite attaining high SyncNet scores, does not fare well in qualitative evaluations (e.g., user studies) of lip-sync, attributed to the production of blurry results and, occasionally, exaggerated

Table 4: Results of face animation on HDTF and MEAD-Neutral.

| Methods | HDTF / MEAD-Neutral | | | | | | |
| --- | --- | --- | --- | --- | --- | --- | --- |
| | SSIM↑ | CPBD↑ | F-LMD↓ | M-LMD↓ | $\text{Sync}_{conf}$↑ | SSIM-M↑ | CPBD-M↑ |
| MakeItTalk [62] | 0.63/0.74 | 0.19/0.10 | 4.10/3.88 | 4.22/5.41 | 3.07/2.02 | 0.62/0.69 | 0.15/0.06 |
| Wav2Lip [30] | 0.75/0.79 | 0.18/0.12 | 2.01/2.38 | 2.54/2.95 | **5.27/3.99** | 0.68/0.78 | 0.08/0.03 |
| PC-AVS [61] | 0.51/0.51 | 0.23/0.07 | 3.64/4.76 | 3.52/3.91 | 4.16/3.09 | 0.60/0.67 | 0.10/0.05 |
| AVCT [46] | 0.73/0.77 | 0.17/0.10 | 2.85/2.68 | 3.53/4.46 | 3.81/2.56 | 0.70/0.73 | 0.16/0.08 |
| EAMM [14] | 0.59/0.41 | 0.08/0.08 | 4.16/7.39 | 4.19/5.03 | 2.30/1.40 | 0.60/0.71 | 0.13/0.05 |
| StyleTalk [26] | **0.78**/0.79 | 0.23/0.12 | 2.10/2.35 | 2.40/2.80 | 4.17/3.05 | 0.76/0.80 | 0.16/0.09 |
| SadTalker [58] | 0.61/0.73 | 0.21/0.12 | 3.98/3.67 | 3.46/4.09 | 4.05/2.62 | 0.61/0.69 | 0.15/**0.10** |
| StyleSync [7] | 0.77/0.80 | 0.21/0.12 | 1.93/2.22 | 2.36/2.76 | 4.21/3.10 | 0.76/0.80 | 0.17/**0.10** |
| Ground Truth | 1/1 | 0.23/0.20 | 0/0 | 0/0 | 4.52/3.57 | 1/1 | 0.21/0.12 |
| **FaceComposer** | **0.78/0.84** | **0.27/0.14** | **1.84/2.16** | **2.25/2.70** | 4.27/3.12 | **0.78/0.83** | **0.18/0.10** |

Table 5: User study results of different methods on HDTF and MEAD-Neutral for the face animation. *LS*, *VQ*, *OR* stand for user study metrics *LipSync*, *VideoQuality* and *OverallRealness*, respectively.

| Methods | HDTF / MEAD-Neutral | | |
| --- | --- | --- | --- |
| | LS↑ | VQ↑ | OR↑ |
| MakeItTalk | 1.71/2.20 | 1.87/2.38 | 1.44/1.74 |
| Wav2Lip | 2.93/3.33 | 1.02/1.10 | 1.10/1.12 |
| PC-AVS | 2.88/3.20 | 1.97/2.46 | 1.73/1.98 |
| AVCT | 2.04/2.76 | 2.60/2.66 | 2.46/2.62 |
| EAMM | 1.90/2.58 | 1.30/1.78 | 1.62/1.94 |
| StyleTalk | 3.10/3.60 | 3.08/3.00 | 2.44/2.82 |
| SadTalker | 3.22/3.68 | 2.82/2.92 | 1.97/2.42 |
| Ground Truth | 4.34/4.56 | 4.06/4.26 | 4.22/4.40 |
| **FaceComposer** | **3.53/3.96** | **3.38/3.50** | **2.93/3.73** |

lip motions. Our method's confidence score aligns most closely with the ground truth, and our method's M-LMD scores are the best. This indicates that `FaceComposer` achieves precise lip-sync. We provide more analysis regarding quantitative evaluation in the *Supplementary Material*.

Fig. 4 shows the qualitative results. It can be seen that MakeItTalk, AVCT, and EAMM fail to produce accurate mouth movements, while `FaceComposer` achieves accurate lip-sync. Although PC-AVS and Wav2Lip are competitive with `FaceComposer` in terms of lip-sync, they can only generate blurry results. `FaceComposer` produces high-fidelity results in the mouth area. StyleTalk often produces overly smooth lip movements. The lip-sync of SadTalker is slightly inferior to our method. StyleSync occasionally yields jittery and exaggerated motions.

We conduct a user study involving 20 participants. Each participant is requested to assign scores to five videos sampled from the test dataset for each method, evaluating them on three aspects: (1) Lip sync accuracy; (2) Video quality; (3) Overall Realness. The scoring employs the Mean Opinion Scores (MOS) rating protocol (scoring from 1 to 5, larger is better). Tab. 5 shows the results. Our method achieves the best in all aspects, which demonstrates the effectiveness of our method.

### 4.3 Advantage of unified framework

In this subsection, we demonstrate the compositional generation capabilities, which are depicted in Tab. 1, of `FaceComposer` with one-time forward. Here we provide two examples of versatile creations. In practical application, one can customize different conditions to meet specific requirements. More results will be provided in *Supplementary Material*.

**Face generation+animation.** `FaceComposer` has the ability of generating a desired face with a random T2F Embedding and animating it with a one-time forward. We show six examples in Fig. 5 to demonstrate such an ability. As we can see, the generated results in each row share almost the same mouth shape as that of the driving PNCCs. Meanwhile, each generated result consistently adheres to the given T2F Embeddings.

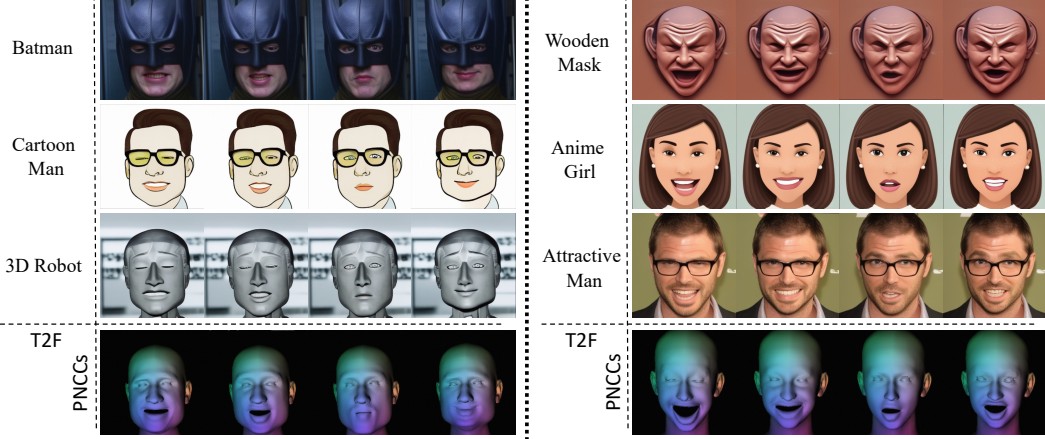

Figure 5: Face generation and animation results. `FaceComposer` supports generating desired faces with T2F Embedding and animating faces with PNCCs simultaneously in a one-time forward.

**Face generation+editing+animation.** Besides, `FaceComposer` can incorporate additional identity conditions to animate a given face in a prescribed style. Some results are shown in Fig. 7. It is clear that each animated frame has the same mouth shape and expression as that of the corresponding PNCCs, while maintaining its original identity and face shape.

## 4.4 Ablation study

To isolate the contributions of our designs, we conduct the ablation studies from three aspects. Note that the Audio2PNCC model is identical in all variants and we only compare the ability of these variants to convert PNCCs to videos.

**Image/Video dataset impact.** We compare the following three variants: (1) only train `FaceComposer` using videos (*w/o* **Image**), (2) only train `FaceComposer` using images (*w/o* **Video**), (3) our full model (**Full**), for face animation on HDTF dataset in Tab. 6. **Full** achieves the best performance in most metrics, which demonstrates the effectiveness of joint training on image and video data. We observe that *w/o* **Video** attains the highest score in CPBD metric. This may be due to two reasons. First, high-fidelity face images from CelebHQ and FFHQ are incorporated into our image data. Second, although our video dataset is in high definition, the motion in the videos may blur certain frames, which may lower the model's score in the CPBD metric.

Table 6: Ablation results of face animation on HDTF

| Method | SSIM↑ | CPBD↑ | F-LMD↓ | M-LMD↓ |
|---|---|---|---|---|
| *w/o* Image | **0.78** | 0.27 | 1.88 | 2.46 |
| *w/o* Video | 0.77 | **0.29** | 1.94 | 2.40 |
| **Full** | **0.78** | 0.27 | **1.84** | **2.25** |

Fig. 6 shows the qualitative ablation results. *w/o* **Image** fails to generate realistic texture in the mouth area, *i.e.* the teeth and the tongue. This may imply that high-resolution image data are crucial to learning to generate high-quality textures. *w/o* **Video** fails to generate mouth shapes corresponding to input PNCCs. This

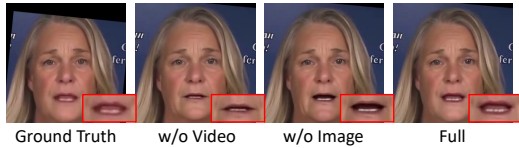

Figure 6: Qualitative ablation results.

may stem from the fact that the image data only contains images with a limited number of mouth shapes. On the other hand, **Full** generates high-fidelity results in the mouth area, produces accurate mouth shape, and attains satisfactory frame consistency. This indicates the necessity of constructing and learning from a multi-modal face database.

**Condition numbers.** We investigate the effect of different numbers of conditions on `FaceComposer` in Tab. 7. Considering face generation/editing/animation are the fundamental tasks, we take T2F, Mask and PNCC as three basic conditions (i.e., **FaceComposer** *w/o* (**ID+S**)), add Sketch condition in **FaceComposer** *w/o* **ID**, and keeps all five conditions in **FaceComposer**. It can be seen that

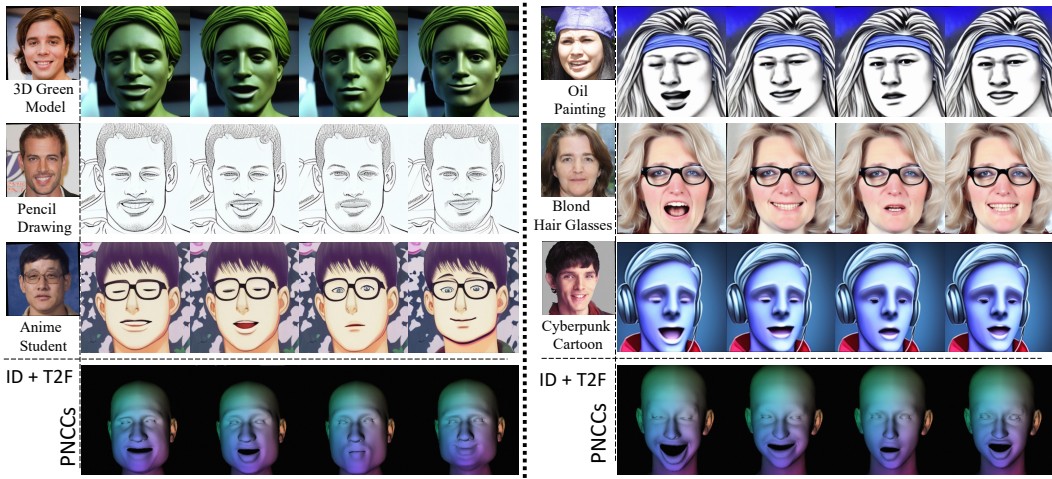

Figure 7: Face generation+editing+animation results. `FaceComposer` is able to generate face images with specified identities, edit the image styles, and animate face images simultaneously in a one-time forward.

Table 7: Ablation results with different conditions or data scales for face generation, editing, and animation.

| Variants | Face Generation | | Face Editing | Face Animation | | | |
| --- | --- | --- | --- | --- | --- | --- | --- |
| | FID↓ | R-precision(%)↑ | IDS↑ | SSIM↑ | CPBD↑ | F-LMD↓ | M-LMD↓ |
| **FaceComposer** | 11.34 | 86.63 | 0.94 | 0.78 | 0.27 | 1.84 | 2.25 |
| FaceComposer *w/o* ID | 11.40 | 86.28 | 0.94 | 0.78 | 0.26 | 1.89 | 2.28 |
| FaceComposer *w/o* (ID+S) | 11.27 | 86.91 | 0.93 | 0.78 | 0.26 | 1.88 | 2.28 |
| FaceComposer- | 11.60 | 86.85 | 0.93 | 0.78 | 0.26 | 1.91 | 2.37 |

FaceComposers with different number of conditions keep stable performance, no matter in face generation, editing or animation task. This is reasonable, bacause the training dataset is fixed, when the number of conditions increases, no additional information is introduced for a specific task.

**Dataset scale.** To demonstrate the impact of dataset scale on generation quality, we show an ablation study in Tab. 7. Considering the state-of-the-art methods generally have dozens of hours of training data, we reduce the training data of `FaceComposer` to the similar scale for a fair comparison. Specifically, we randomly sampled 10 hours of video and 4.5W images from our original dataset to train a `FaceComposer` (denoted as FaceComposer-). From Tab. 7, it can be observed that FaceComposer- is inferior to `FaceComposer` due to the decrease of data information, but it is still better than other state-of-the-arts in Tabs. 2 to 4.

## 5   Conclusion

This paper introduces a unified framework (named as `FaceComposer`) for versatile facial content creation, *i.e.*, 1) generating faces from scratch, 2) adjusting portion of the generated ones, and 3) making the generated target move. It demonstrates a complete creative process from static (image) to dynamic (video) content. `FaceComposer` employs a LDM with multiple conditions to handle all above facial creation tasks. We evaluate its performance in various tasks and demonstrate our versatile creative ability in different combined tasks. Demo interface, Social impact and Ethic statement will be moved to *Supplementary Material* due to limited space.

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
