# FaceComposer: A Unified Model for Versatile Facial Content Creation
## $<$*Supplementary Material* $>$

In this Appendix, we show training and data details in parts A, B, and C. Part D demonstrates more experimental results, followed by Part E, the demo page introduction. At last, we list the limitations and potential societal impact of our `FaceComposer`.

## A    Architecture details.

|  | Prior | LDM |
|---|---|---|
| Diffusion steps | 1000 | 1000 |
| Noise schedule | cosine | cosine |
| Sampling steps | 100 | 50 |
| Sampling variance method | dpm-solver | dpm-solver |
| Model size | 1B | 1B |
| Channels | - | 320 |
| Depth | - | 2 |
| Channels multiple | - | 1,2,4,4 |
| Heads channels | - | 64 |
| Attention resolution | - | 32,16,8 |
| Dropout | - | 0.1 |
| Weight decay | 6.0e-2 | - |
| Batch size | 4096 | 1024 |
| Iterations | 1M | 1M |
| Learning rate | 1.1e-4 | 5e-5 |
| Adam $\beta_2$ | 0.96 | 0.999 |
| Adam $\epsilon$ | 1.0e-6 | 1.0e-8 |
| EMA decay | 0.9999 | 0.9999 |

Table A1: Hyperparameters for `FaceComposer`. We use DPM-Solver++ [8] as the sampling algorithm for all diffusion models.

The hyperparameters of `FaceComposer` are reported in Tab. A1. The Conditioning mechanism is depicted in Fig. A1 (global conditioning) and Fig. A2 (local conditioning). For guidance direction in all our tasks, we put conditions to be emphasized into subset $c_2$, set $c_1$ empty. One exception is local conditions, which will be added into $c_1$ too, since we observed that this operation helped `FaceComposer` generate better results.

The architecture of the Audio2PNCC model follows that of StyleTalk [9]. Unlike StyleTalk, We change the input audio representations from phonemes to Wav2Vec [1] features. Besides, the Audio2PNCC model predicts FLAME parameters instead of BFM parameters used in StyleTalk. The Audio2PNCC model predicts 50 expression parameters and 3 jaw pose parameters. The predicted parameters are merged with other parameters (*e.g.* shape, global pose, and camera parameters) to render the PNCC images [14].

37th Conference on Neural Information Processing Systems (NeurIPS 2023).

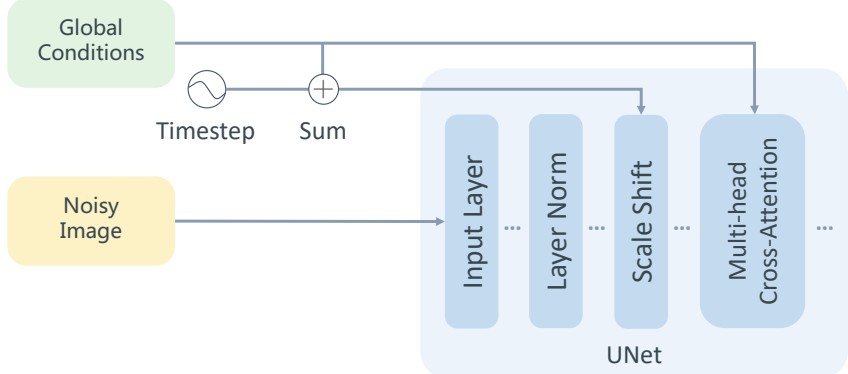

Figure A1: Global conditioning mechanism of `FaceComposer`. Global conditions include Identity Feature and T2F Embedding. For Identity Feature, we project and add it to the timestep embedding. For T2F Embedding, we take it as key and value for multi-head cross-attention modules.

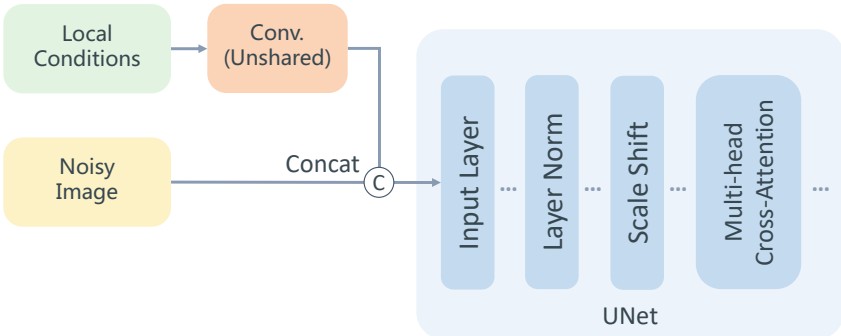

Figure A2: Local conditioning mechanism of `FaceComposer`. Local conditions consist of Mask, PNCC and Sketch. We project these conditions into embeddings with the same spatial size as the noisy image using stacked convolutional layers. Subsequently, we concatenate the embeddings and the noisy image as the input of UNet.

## B    Training details.

During the training stage, our model starts from a pre-trained LDMs[1], and is further trained on our multi-modal face database through a joint training mechanism. We apply the Adam optimizer with $\beta_1 = 0.9$, $\beta_2 = 0.999$, and an initial learning rate of $5 \times 10^{-5}$ for the LDMs training. The prior model is trained with another Adam optimizer with $\beta_1 = 0.9$, $\beta_2 = 0.96$, and an initial learning rate of $1.1 \times 10^{-4}$. In total, we pretrain the LDMs with $1M$ steps on the full multi-modal dataset using only T2F Embeddings as the condition, and then finetune the model for 200K steps with all conditions enabled. The total training costs approximately four days with eight NIVIDA A100 GPUs. We use an independent dropout probability of $0.5$ for each condition, a probability of $0.1$ for dropping all conditions, and a probability of $0.1$ for retaining all conditions. The prior model is trained for $1M$ steps on the image dataset. In terms of joint training mechanism, we allocate half of GPUs to perform image training, while the rest of the GPUs are dedicated to video training.

To train the Audio2PNCC model, we follow the settings in StyleTalk [9]. We train on HDTF for 12 epochs. It takes 0.5 hours to train the model on 1 A100 GPU.

## C    Details of the multi-modal face database

### C.1    Image data collection and pre-processing

We merge LAION-Face [13], FFHQ [7], and CelebHQ [6] to construct the raw data pool of image data. In order to sample high-quality image-text pairs from the raw data pool, we clean up LAION-

---

[1]https://github.com/Stability-AI/stablediffusion

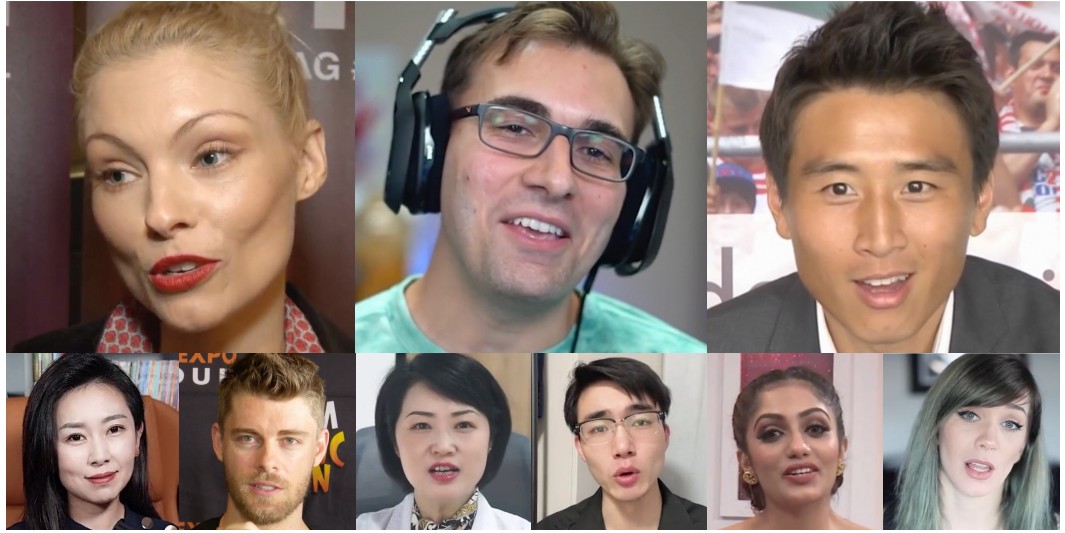

Figure A3: Samples from the video part of the multi-modal face database.

Face. First, we use Face CLIP [13] to filter out the image-text pairs whose text descriptions do not match the images. Specifically, for each image-text pair, we compute the cosine similarity between CLIP features extracted from the image and the text and filter out the pair if the similarity is lower than a predefined threshold 0.3. Second, we use an off-the-shelf face detector [3] to detect faces in images and filter out images with no faces detected. Third, we detect the bounding boxes of faces and remove images in which the size of the bounding box is lower than $256 \times 256$. Finally, we obtain the cleaned LAION-Face dataset. It contains 1 million face images with corresponding text captions. After incorporating images from CelebHQ and FFHQ, the database contains 1.1 million text-annotated facial images.

## C.2    Video data collection and preprocessing

To construct the raw data pool of video data, we create various queries and use these queries to collect raw videos from the Internet. In order to sample high-quality talking head video clips from the raw data pool, we design several constraints to filter out unsatisfactory videos. First, we split the videos into 5-second clips and check whether all the clip frames belong to the same person based on the identity (detected by an identity recognition network [2]). If not, the clips are filtered out. Second, when multiple faces appear in a clip, we only keep the clips in which the size of one person's face is at least four times larger than those of others. Third, we discard clips with head rotation greater than 45 degrees along any axis. Fourth, we remove the clips with faces in low resolution. The average face size of selected clips is larger than $200 \times 300$ pixels. The total video duration is more than 500 hours. Samples are shown in Fig. A3.

## D    More analysis on the evaluation of face animation

### D.1    Why `FaceComposer` uses masks in face animation evaluations

When generating face animation, `FaceComposer` takes images with the mouth region masked as input and generates images with the mouth region inpainted (visual dubbing methods), rather than directly generating the entire face (one-shot methods). We would like to explain why it's unreasonable to make `FaceComposer` in a one-shot version (which generates the entire face) compete against one-shot talking head methods. In the one-shot setting, the one-shot image can be input into one of the four conditions (i.e. Mask, Sketch, ID and T2F. PNCCs are used for facial motions). When using a one-shot image for Mask, no region should be masked to make it the same as one-shot talking head methods. But this use of `FaceComposer` does not match its training scheme, where we mask one/all of the nine face parsing areas and recover the masked region based on other conditions. Worse, the facial motion in unmasked images may conflict with that in PNCCs, resulting in undesired face

motion generation. As for the other three conditions (Sketch cannot capture the textual information in the one-shot image, ID can only capture identity attributes without any pixel-level information, T2F is only able to capture global information), the illumination or speaker appearance of the generated videos may differ from the input images, and all of them are not designed for one-shot talking head scenario.

## D.2 Fair comparison with prior arts

How to conduct a fair comparison between visual dubbing methods and one-shot methods is neglected in prior arts. We may attribute the unfairness issue to two aspects: (1) it is unfair to compare the quality in the non-mouth area, as visual dubbing methods "copy" the area from the input images while one-shot methods generate the area by themselves. (2) the pose of the generated speaker in one-shot methods may change, making it fall short in metrics where ground truth (GT) is used as a reference. To address these issues, we introduce new metrics CPBD-M and SSIM-M.

For (1), CPBD-M, SSIM-M, and M-LMD all focus on the video quality of the mouth region. For (2), if the pose of visual dubbing and one-shot methods are both aligned with GT, we argue that, in addition to mouth movement accuracy, the video sharpness and appearance consistency (whether the appearance is consistent with that in the input image) of the mouth area should also be evaluated quantitatively, as the quality of these two properties is crucial to video realness. To get closer to this goal, the samples we generated use the head pose and neutral emotion of GT for pose/emotion-controllable methods, such as PC-AVS, StyleTalk, and EAMM. (We acknowledge for methods that can not control poses, unfairness may still exist.) Besides, CPBD-M is a no-reference metric and SSIM-M evaluates structural similarity but not pixel-level one, both can mitigate the effects of speaker pose changes.

## D.3 Why SyncNet score is lower than Wav2Lip

The SyncNet score of `FaceComposer` is only lower than Wav2Lip, higher than all other methods, and the closest to GT. Wav2Lip achieves the highest SyncNet score, even higher than GT, since it uses SyncNet as a discriminator. Numerous prior arts [4, 12, 9, 5, 10] have reported that Wav2Lip, despite attaining high SyncNet scores, does not fare well in qualitative evaluations (e.g., user studies) of lip-sync, attributed to the production of blurry results and, occasionally, exaggerated lip motions. StyleSync [4] claims SyncNet score only reflects how well an audio-visual pair fits the learned SyncNet model rather than the true perceptual quality. Thus though generated results might outperform GT on the metric, it does not mean better sync quality. SPACE [5] claims that SyncNet scores are very sensitive to the input crop, PD-FGC [10] claims that the SyncNet Score of a method is strongly correlated with its training data, which makes it unfair when comparing methods trained on different data.

# E More experimental results.

## E.1 Face generation+animation

In the main paper, we have shown the generated results conditioned on PNCC sequence and T2F embedding, here we display another two conditions combination: PNCC sequence and Identity Feature to finish face generation and animation simultaneously. Considering it's a face animation, we put it into the attached video file (suppvideo.mp4), where we get Identity Feature from source images and PNCC sequence from audio.

## E.2 Face generation+editing

We show more face generation+editing results in Fig. A4. For Fig. A4(a), we extract Identity Feature from source images and mask the faces of target images. In this setting, we can achieve the same effect as face swapping, demonstrating our abilities of versatile facial content creation again. For Fig. A4(b), images in the first row provide Identity Feature, the second row lists the style prompt, and the third row shows the generated results.

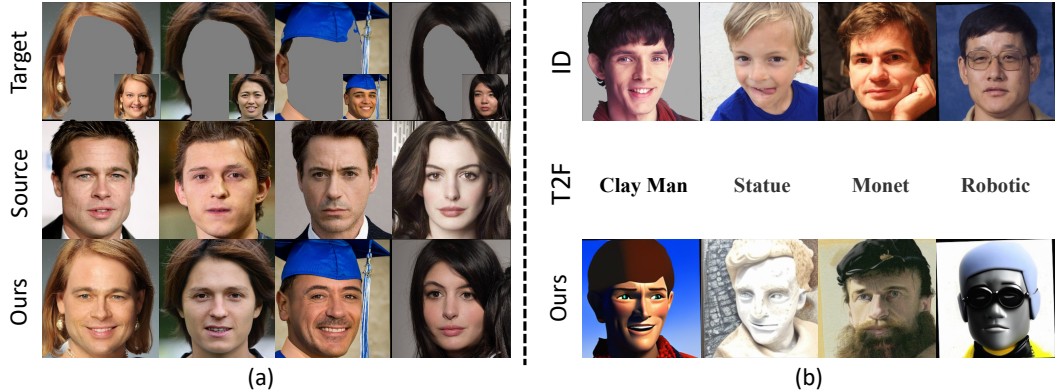

Figure A4: More face generation+editing results. (a) Results are conditioned on Identity Feature and Mask; (b) Identity Feature and T2F Embedding are used as conditions.

### E.3 Face generation+editing+animation

We show some video results, corresponding to Fig. 7 in the main paper, in the final video file (suppvideo.mp4). It can be seen that `FaceComposer` can generate smooth face sequences with high relevance to the conditions.

## F Interface design.

In order to make people create, edit and drive their desired characters with just one-click, we design a friendly interactive interface in Fig. A5. Thanks to the publicly available code CoAdapter [2], we build a demo of `FaceComposer` based on the open-source tool Gradio [3].

The five rectangular boxes in the first row represent five condition inputs: Mask, PNCC, Sketch, ID and T2F from left to right. `FaceComposer` supports condition combinations, so you can check "nothing" in the boxes for the conditions you don't need, and drop image/video to the corresponding boxes or click that boxes to upload them for the conditions you want. For Mask, we will provide nine parsing choices to help you mask the uploaded image or video. For PNCC, one can drop an audio to the box, then the audio2PNCC module will be called to extract the PNCC sequence. For Sketch, we will automatically extract the sketch for the uploaded image or video. For ID, only reference image is needed, ArcFace model will be called to get the ID. For T2F, you can upload the reference image, where T2F is obtained from Face Clip model, or use the input box for "Prompt / Negative Prompt" below to make the T2F extracted from the text. The negative prompt will be put into $c_1$, and prompt is in $c_2$. Below the input box of the prompt, we show five parameters: (1) Guidance Scale is the guidance weight we used in guidance directions; (2) Num samples mean how many samples you want to generate; (3) Seed is the random seed used in random process in `FaceComposer`, like noise generation; (4) Steps mean the number of DDIM steps in inference process; (5) Image resolution, just as its name implies.

## G Discussion.

**Pros and cons.** The analysis of the pros and cons of jointly training with different conditions is listed below: **Pros:** Training with multi-conditions, `FaceComposer` can support different tasks (e.g. face generation, face editing and face animation) and enjoy diverse controllabilities (e.g. accomplishing combined tasks among face creating, editing and animating with one-time forward) with a unified model. And Tab. 7 in main paper shows that the performance of `FaceComposer` keeps stable when changing the number of conditions on a fixed dataset. **Cons:** Due to the general design of `FaceComposer` for a variety of tasks, we need more training data to make different tasks perform well, which inevitably increases the cost of training. With a limited scale of training data, the performance

---

[2]https://huggingface.co/spaces/Adapter/CoAdapter
[3]https://github.com/gradio-app/gradio

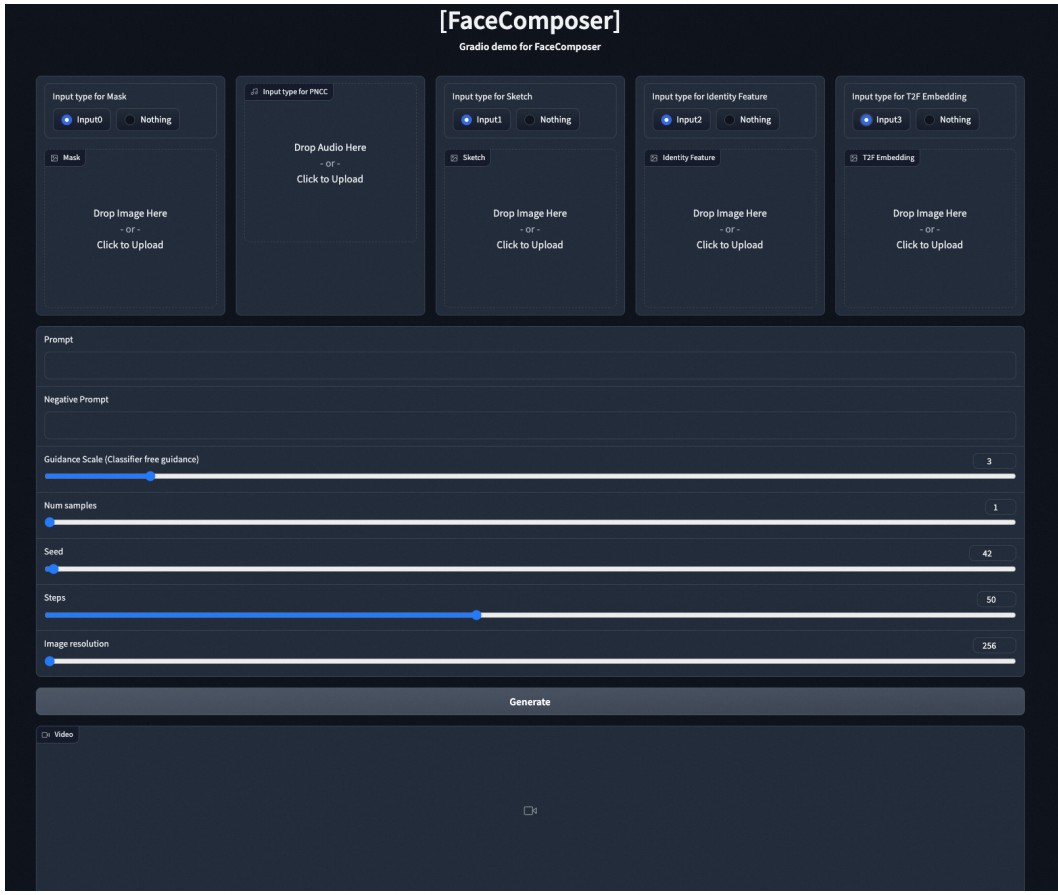

Figure A5: The demo page of `FaceComposer`.

of `FaceComposer` may degrade on some tasks, as illustrated in Tab. 7 of main paper. We thus collect a high-quality large-scale multi-modal face database to alleviate such a problem.

**Limitations.** Although `FaceComposer` supports versatile facial content creation by combining diverse conditions, we fail to generate the whole human (face and body). Besides hard to make different conditions control different human parts, potential conflicts between conditions and inadequate data lead the "HumanComposer" to a more challenging task. Another limitation of `FaceComposer` is the resolution of generated results, i.e., 256×256, which is a compromise between quality and training efficiency. We will increase the resolution to 512×512 when the code is released in the near future.

**Potential societal impact.** Since `FaceComposer` integrates various facial creation tools within a unified interface, which facilitates access to each tool and enables their combined utilization, the potential downstream applications of `FaceComposer` are diverse and may have complex societal effects. On the one hand, `FaceComposer` shows considerable promise in enhancing, extending, and complementing human creativity. Besides, `FaceComposer` may lead to the creation of new tools for creative practitioners, allowing for an expansion of existing options. On the other hand, `FaceComposer` can be utilized for malevolent purposes. It is possible for `FaceComposer` to produce content containing or suggesting sexual content, hatred, or violence. `FaceComposer` may cause detrimental effects on persons and communities when `FaceComposer` is used to erase or denigrate them, reinforce stereotypes, subject them to indignity, or provide them with disparately low-quality performance. Harassment, bullying, or exploitation of individuals is another possible abuse. `FaceComposer` also could be exploited to deceive or misinform individuals. As `FaceComposer` makes it easier to generate content, these negative impacts will be exacerbated if there are no countermeasures in place.

Numerous precautions have been or will be taken before the release to prevent the misuse of `FaceComposer`. We remove harmful content in the training dataset and involve prompt and image/video filters to prevent users from generating harmful content. We will improve the representa-

tiveness of the dataset by manually balancing it to avoid the bias of generation results, which will mitigate erasure, stereotype reinforcing, indignity, and disparately low-quality performance for some prompts. To prevent harassment and bullying, we will ask users not to upload images of people without their consent. To avoid `FaceComposer` being used to misinform individuals, all results generated by `FaceComposer` will be marked as synthetic content using watermarks. We will perform pre-release risk analysis by utilizing an expanding array of safety evaluations and red teaming tools. We will also examine the outcomes of pilots involving novel use cases and carry out comprehensive retrospective reviews. Automated and human monitoring systems will be developed in order to prevent the occurrence of misuse. We will do our best to keep researching to eliminate negative social impacts.

Collecting and releasing a large-scale talking head dataset may also engender multifaceted societal implications. In addressing the ethical considerations of our dataset, we align our practices with those established by TalkingHead-1KH [11], a public video dataset. Similar to TalkingHead-1KH, our dataset comprises exclusively video clips under permissive licenses, such as the Creative Commons BY 3.0 license, which permits reuse. When releasing our dataset, we will only provide the original video URL, rather than the video content itself, allowing content owners to retain control over their videos. If a content owner wishes to remove a video from our dataset, they can either modify the license of their video and remove it from the original URL or directly inform us via email to delete the video URL; the email will be provided upon the release of the dataset.

We have invested significant time and effort in meticulously filtering the dataset using off-the-shelf tools like face detection, pose estimation, and face identification before employing five laborers to select high-quality data. These laborers were compensated fairly, based on their workload, with prices negotiated in advance, acknowledging the labor involved in the collection of the dataset.

Given that our dataset is crawled from the Internet, it inherently embodies the biases from the Internet. We are actively taking measures to enhance the dataset's representativeness by incorporating more data representing minority races/ethnicities and reducing data from majority races/ethnicities. We will also provide a data augmentation module to those utilizing our dataset. Despite our endeavors to balance the dataset, issues regarding representativeness may persist. We plan to alleviate this problem over time as we gather more data and feedback.

To regulate the usage, the model and dataset can only be obtained upon email request for research purposes only. The requesters will be asked to specify their full name, their institutes, and their positions and to adhere to our usage policies.