# OpenReview forum: "FaceComposer: A Unified Model for Versatile Facial Content Creation"
_NeurIPS.cc/2023/Conference — NeurIPS 2023 poster_

### Official Review · Reviewer_UEp9 · 2023-06-13

**Soundness:** 4 excellent
**Presentation:** 4 excellent
**Contribution:** 4 excellent
**Rating:** 7
**Confidence:** 4

**Summary:**

The advancement of generative models has significant progress in automatic facial content creation. However, current models pose challenges due to their high customization and inefficiencies. To address this, a unified model called FaceComposer is proposed.

- The model leverages images, videos, multi-modal face datasets.
- Better performance than SOTA.
- The user-friendly interface of FaceComposer enables easy face generation, editing, and animation.


**Strengths:**

- Well written text with nice figures and numerical results.
- Promising results supported by numerical results.
- Proposed method works in many styles
- Unified framework enables the proposed method to work on multiple tasks.


**Weaknesses:**

- The interface is confusing.
- Maybe adding human eval can make the work better.



**Questions:**

The framework is able to deal with various tasks by using different forms of datasets. I would like to know how much data and computation cost we need for each task in order to make the proposed method work.

**Limitations:**

- Excellent method supported by well designed experiments.
- Maybe adding human eval can make the work better.

---

> ### Author Rebuttal · Authors · 2023-08-09
>
> Thank you for your comments!
>
> $\textbf{1. Confusing interface.}$
>
> We apologize for the confusion caused by the user interface. Here, we will provide a detailed explanation of the demo page in Figure A6. The five rectangular boxes in the first row represent five condition inputs: Mask, PNCC, Sketch, Identity Feature (IF) and T2F Embedding (T2F) from left to right. FaceComposer supports condition combinations, so you can check "nothing" in the boxes for the conditions you don't need, and drop image/video to the corresponding boxes or click that boxes to upload them for the conditions you want. For Mask, we will provide nine parsing choices to help you mask the uploaded image or video. For PNCC, one can drop an audio to the box, then the audio2PNCC module will be called to extract the PNCC sequence. For Sketch, we will automatically extract the sketch for the uploaded image or video. For IF, only reference image is needed, ArcFace model will be called to get the IF. For T2F, you can upload the reference image, where T2F is obtained from Face Clip model, or use the input box for "Prompt / Negative Prompt" below to make the T2F extracted from the text. The negative prompt will be put into $c_1$, and prompt is in $c_2$ (the explanation of $c_1$ and $c_2$ can be found in line 107~109 of the main paper). Below the input box of the prompt, we show five parameters: (1) Guidance Scale is the guidance weight we used in guidance directions; (2) Num samples mean how many samples you want to generate; (3) Seed is the random seed used in random process in FaceComposer, like noise generation; (4) Steps mean the number of DDIM steps in inference process; (5) Image resolution, just as its name implies. We hope these words clear your confusion and we will consider refine the interface in the final version to make it easy to follow.
>
> $\textbf{2. Human eval.}$
>
> Thanks for your suggestion. Table R2 and R3 demonstrate the user studies we add for face generation and face animation, we skip face editing task since its user study has been shown in Table 3 of the main paper. Participants are asked to score twenty images sampled from testset for each SOTA in Table R3 and ten videos in Table R2, where Accuracy and Realism keep the same meaning as we claim in line 195~196 of the main paper, LipSync is the sync degree between lip movements and speech content, OveralRealness tells which video is more natural and realistic, VideoQuality depicts the visual quality of generated video. It can be observed that FaceComposer achieves the best results in terms of both face generation and face animation.
>
> $\textbf{3. Data and computation cost.}$
>
> We apologize for the misunderstanding caused by our unclear description. We only have one training data: 1.1M images + 500 hours videos, and do not use different datasets for different tasks. We decompose this multi-modal database into five face-specific conditions to train a unified model. Then in the inference phase, we can make different combinations of these conditions to achieve different tasks, as we list in Table 1 of the main paper. We spend approximately four days training FaceComposer with eight NIVIDA A100 GPUs, based on a pre-trained Latent Diffusion Models.

---

> > ### Comment · Reviewer_UEp9 · 2023-08-18
> >
> > Thanks for the comments. I read them on day one. I will keep my original rating.
> > Nice work

---

> > > ### Author Response · Authors · 2023-08-21
> > >
> > > We really appreciate your constructive feedback to improve our work, thanks!
> > >
> > > Best regards,
> > >
> > > The Authors

---

### Official Review · Reviewer_BHq7 · 2023-06-30

**Soundness:** 3 good
**Presentation:** 3 good
**Contribution:** 1 poor
**Rating:** 5
**Confidence:** 4

**Summary:**

The paper presents an all-in-one pipeline that can perform face generation, editing and animation and can be driven by multiple signals such as audio, text and sketches. The proposed model is based on a Latent Diffusion Model (LDM) and works by decomposing images faces into several representations capturing identity (ArcFace embeddings), geometry (PNCCs), face shape (sketch embeddings) and text description (T2F embeddings). These embeddings are then used as conditions and guidance for the LDM, which is taken pre-trained and fine-tuned on a large dataset containing both images and videos. The model is able to train simultaneously with images and videos thanks to temporal attention modules inserted in the U-Net architecture. The model is evaluated on face generation, editing and animation and compared to SoTA methods for each task. Facial animation is performed using an audio-to-PNCC model, which predicts the 50 expression parameters and 3 jaw pose parameters from Wav2Vec embeddings.

**Strengths:**

The proposed framework is indeed flexible and can perform tasks that would normally require multiple modules. The advantage of a unified model apart from the simplicity is the computational efficiency (i.e. low latency), since it can perform several steps in parallel without stringing together multiple models.

The authors have collected a curated, extensive dataset of images and videos. They make sure that the dataset only contains high quality frames and highly correlated text-image pairs. The dataset will be made public and will undoubtedly be very useful for researchers in the field.

The authors include an ablation study showing the effect of training on only videos or only images and share their hypotheses into why the addition of video data results in slightly blurrier frames.

The lip movements appear to be synchronised with the audio and this is also visible in the SyncNet and LMD metrics.

**Weaknesses:**

The proposed framework does not present significant methodological novelty and it is not clear what the key contributions are besides the collection of multiple conditions and pretrained models to drive a unified framework. Temporal attention has been used before in [*] and [**] and the conditioning mechanism and guidance used in the paper are standard practice for most diffusion models. Make-a-Video [*] also is trained on both image and video data and has a similar approach but is missing from the references. The audio-to-PNCC network is based on StyleTalk but is only described in the supplementary material.

From the supplementary video the performance of the model seems to degrade a lot when the masks are not used (identity feature + PNCCs). This setting is much more challenging and the proposed model seems to struggle to preserve the identity or produce natural head motion. Furthermore the visual quality also seems to degrade. The authors also do not mention if the results of Table 4 are measured when using masks or not. If masks are used then this is an unfair comparison against other methods such as PC-AVS that are truly one-shot and do not simply in-paint the mouth. This would also explain why the SSIM and CPBD are better.

The authors only have an ablation study examining how the use of images and videos during training affects the performance of the model. They have not performed ablation studies to explore how the performance improves with the addition of each condition.

The description of the user interface of the tool is not of particular interest for machine learning research. The authors should consider removing this section from the main paper and using the space to add experiments such as ablation studies or to describe some of the components in more detail.

[*] Singer et.al. "Make-A-Video: Text-to-Video Generation without Text-Video Data"
[**] Ho et. al. "Video Diffusion Models"

**Questions:**

The initial clips in the supplementary material seem to be of a much higher quality than the ones that follow. The authors should clarify if the masks are used for the initial videos. This is will help to put the performance of the model in perspective and determine if it should be judged as a video dubbing or a one-shot speech-driven facial animation method.

How does the temporal attention used in this paper differ from that used in Make-A-Video[*] or Video Diffusion Models[**]

[*] Singer et.al. "Make-A-Video: Text-to-Video Generation without Text-Video Data"
[**] Ho et. al. "Video Diffusion Models"

**Limitations:**

The authors discuss the limitations of the method in the supplementary material. They also point out some negative applications of methods such as the one proposed in the paper.

---

> ### Author Rebuttal · Authors · 2023-08-09
>
> Thank you for your comments!
>
> $\textbf{1. Novelty and contributions.}$
>
> We would like to clarify our primary contribution, which is the unified generative framework with various means of controllability for versatile facial content creation. Our design enjoys some merits in both training and inference stages. For training, we only need to train and save one model for most tasks, reducing memory and computation cost. For inference, multi-condition-driven scheme enables us to accomplish combined tasks in one-time forward, improving inference flexibility and efficiency. We also collect a high-quality multi-modal face database to ensure the success of FaceComposer, and we hope such a database would be beneficial to further studies in this field. Besides, thanks for the reminder of the missing references: Make-A-Video and Video Diffusion Models. We will add more discussion in the revision.
>
> $\textbf{2. Mask influence.}$
>
> Firstly, we agree with you that the setting, i.e. Identity Feature (IF) + PNCCs, is challenging since IF only indicates the identity attribute and PNCCs only represent facial geometric information, both of which do not contain detailed information, like texture, skin or hair color.
>
> Secondly, FaceComposer in Table 4 and initial clips in supplementary material have mask, and we show the mask shape in Figure R1. We agree with you that there may exist an unfair comparison between video dubbing (like Facecomposer, Wav2Lip) and one-shot talking head (like PC-AVS), although this setting is very **common** in many talking head practices [1][2][3]. Considering that both categories of methods need to generate the mouth, we add evaluation metrics only for the mouth region (SSIM-M and CPBD-M, besides M-LMD) to reduce the influence of mask, as shown in Table R1. It can be seen our SSIM-M and CPBD-M are still the best, proving the effectiveness of FaceComposer in non-mask areas.
>
> Thirdly, we want to claim the mask is not the key to the generation quality of FaceComposer, and we justify it by given more videos (please see Rebuttal_Video.mp4 submitted to AC), including results of IF + PNCCs and other condition combinations without mask. The high generation quality could be attributed to the foundation generative model (LDM) and the high-quality training data.
>
> [1] A lip sync expert is all you need for speech to lip generation in the wild, ACMMM 2020
> [2] Masked Lip-Sync Prediction by Audio-Visual Contextual Exploitation in Transformers, SIGGRAPH Asia 2022
> [3] StyleSync: High-Fidelity Generalized and Personalized Lip Sync in Style-based Generator, CVPR 2023
>
> $\textbf{3. Ablation studies of condition impact.}$
>
> Following your suggestion, we conduct ablation studies in Table R4 (upper half) to explore the condition impact. We want to explain the performance is stable, not improved, with the addition of each condition, because we only increased the number of conditions decomposed from the training data, not the training data itself. Taking the face generation in Table R4 (upper half) as example, adding extra conditions (e.g. sketch, IF) into baseline will not bring additional benefits, since they only use the T2F Embedding as condition in the test phase. In contrast, increasing the size of dataset will introduce new data information, which can improve the performance as we show in Table R4 (bottom half).
>
> $\textbf{4. User interface.}$
>
> Thanks for the suggestion. We will remove the user interface to Supplementary Material to save more spaces for experiments.
>
> $\textbf{5. Difference of temporal attention.}$
>
> The temporal attention in FaceComposer has no difference with that in Make-A-Video/Video Diffusion Models. Describing this module is to make our paper self-contained. We will add the missing references to avoid misunderstanding.

---

> > ### Comment · Reviewer_BHq7 · 2023-08-17
> >
> > Thank you for the explanations and additional experiments. I believe that they make the paper clearer for the reader and answered many of my questions. I still believe that the proposed method falls short in facial animation and that the comparison with one-shot approaches is not fair. Since the model is flexible enough to perform one-shot facial animation I do not see the reason why a fair comparison can't be made in this case (i.e., use the masked video version to compare to dubbing approaches and the one-shot version for the others). The only thing that can be directly compared between dubbing and one-shot approaches is mouth movement accuracy. Unfortunately, 2 of the 3 metrics added do not reflect the accuracy of the mouth movement. CPBD and SSIM do not capture mouth movement accuracy and LMD is dependent on mouth position so it may penalise methods capable of generating novel emotions or head movements. SyncNet score is the best metric for mouth accuracy but the proposed approach does not outperform Wav2Lip.
> >
> > I have also noticed something strange in the reported LMD of Wav2Lip. The F-LMD seems to be higher than M-LMD. This would be understandable for one shot methods but for Wav2Lip (which only inpaints the mouth) I would expect F-LMD to be lower than M-LMD (since landmarks of the rest of the face will match with those of the ground truth).

---

> > > ### Author Response · Authors · 2023-08-19
> > >
> > > Thanks for your insightful comments.
> > >
> > > $\textbf{1. Why FaceComposer uses masks in face animation evaluations}$
> > >
> > > Firstly, we would like to explain why it's unreasonable to make FaceComposer in one-shot version compete against one-shot talking head methods. In the one-shot setting, the one-shot image can be input into one of the four conditions (i.e. Mask, Sketch, IF and T2F. PNCCs is used for facial motions). When using one-shot image for Mask, no region should be masked to make it the same as one-shot talking head methods. But this use of FaceComposer does not match its training scheme, where we mask one/all of nine face parsing areas and recover the masked region based on other conditions. Worse, the facial motion in unmasked image may conflict with that in PNCCs, resulting in undesired face motion generation. As for the other three conditions (Sketch cannot capture the textual information in one-shot image, IF can only capture identity attribute without any pixel-level information, T2F is only able to capture global information), the illumination or speaker appearance of the generated videos may differ from the input images, and all of them are not designed for one-shot talking head scenario.
> > >
> > > Secondly, to further validate the effectiveness of our method, we add a comparison with StyleSync [1], a recently published visual dubbing method whose generation setting is the same as that of FaceComposer (in-painting masked mouth area). As shown in the table below, our method achieves better performance in all metrics, indicating the superiority of FaceComposer.
> > >
> > > $\textbf{2. Fair comparisons with prior arts.}$
> > >
> > > It is very insightful for the reviewer to consider how to conduct a fair comparison between visual dubbing methods and one-shot methods, which is neglected in prior arts. We may attribute the unfairness issue to two aspects: (1) it is unfair to compare the quality in the non-mouth area, as visual dubbing methods "copy" the area from the input images while one-shot methods generate the area by themselves. (2) the pose of the generated speaker in one-shot methods may change, making it fall short in metrics where ground truth (GT) is used as reference.
> > >
> > > For (1), CPBD-M, SSIM-M, and M-LMD all focus on the video quality of the mouth region. For (2), if the pose of visual dubbing and one-shot methods are both aligned with GT, we argue that, in addition to mouth movement accuracy, the video sharpness and appearance consistency (whether the appearance is consistent with that in the input image) of the mouth area should also be evaluated quantitatively, as the quality of these two properties is crucial to video realness. To get closer to this goal, the samples we generated in Table R1/R2 use the head pose and neutral emotion of GT for pose/emotion-controllable methods, such as PC-AVS, StyleTalk, and EAMM. (We acknowledge for methods that can not control poses, unfairness may still exist.) Besides, CPBD-M is a no-reference metric and SSIM-M evaluates structural similarity but not pixel-level one, both can mitigate the effects of speaker pose changes.

---

> > > > ### Author Response · Authors · 2023-08-19
> > > >
> > > > $\textbf{3. Why SyncNet score is lower than Wav2Lip.}$
> > > >
> > > > The SyncNet score of FaceComposer is only lower than Wav2Lip, higher than all other methods, and the closest to GT. Wav2Lip achieves the highest SyncNet score, even higher than GT, since it uses SyncNet as a discriminator. Numerous prior arts [1][4][5][6][7] report and analyze that Wav2Lip with a high SyncNet score does not perform well in the qualitative evaluation (e.g. user study) of lip-sync due to the blurry results and sometimes exaggerated lip motions. StyleSync [1] claims SyncNet score only reflects how well an audio-visual pair fits the learned SyncNet model rather than the true perceptual quality. Thus though generated results might outperform GT on the metric, it does not mean better sync quality. SPACE [6] claims that SyncNet scores are very sensitive to the input crop, PD-FGC [7] claims that the SyncNet Score of a method is strongly correlated with its training data, which makes it unfair when comparing methods trained on different data.
> > > >
> > > > $\textbf{4. Wav2Lip LMD scores.}$
> > > >
> > > > This phenomenon stems from the fact that Wav2Lip samples are generated using the first image as the reference (following the practices in [8][5][9]), which means the upper faces in generated videos are static. Since speakers in HDTF exhibit large head movements, F-LMD is large. To address your concern, we generate Wav2Lip samples using the whole video as a reference and report the results in the following table.
> > > >
> > > > $\textbf{5. Results.}$
> > > >
> > > >
> > > > |  MEAD                      | SSIM $\uparrow$ | CPBD $\uparrow$ | F-LMD $\downarrow$ | M-LMD $\downarrow$ | Sync_conf $\uparrow$ | SSIM-M $\uparrow$ | CPBD-M $\uparrow$ |
> > > > |---------------------------|------|------|-------|-------|-----------|------------|------------|
> > > > | Wav2Lip (video reference) | 0.79 | 0.12 | 2.38  | 2.95  | 3.99      | 0.78       | 0.03       |
> > > > | StyleSync                 | 0.80 | 0.12 | 2.22  | 2.76  | 3.10      | 0.80       | 0.10       |
> > > > | FaceComposer              | 0.84 | 0.14 | 2.16  | 2.70  | 3.12      | 0.83       | 0.10       |
> > > >
> > > > |     HDTF                      | SSIM $\uparrow$ | CPBD $\uparrow$ | F-LMD $\downarrow$ | M-LMD $\downarrow$ | Sync_conf $\uparrow$ | SSIM-M $\uparrow$ | CPBD-M $\uparrow$ |
> > > > |---------------------------|------|------|-------|-------|-----------|------------|------------|
> > > > | Wav2Lip (video reference) | 0.75 | 0.18 | 2.01  | 2.54  | 5.27      | 0.68       | 0.08       |
> > > > | StyleSync                 | 0.77 | 0.21 | 1.93  | 2.36  | 4.21      | 0.76       | 0.17       |
> > > > | FaceComposer              | 0.78 | 0.27 | 1.84  | 2.25  | 4.27      | 0.78       | 0.18       |
> > > >
> > > > $\textbf{Reference.}$
> > > >
> > > > [1] StyleSync: High-Fidelity Generalized and Personalized Lip Sync in Style-based Generator, CVPR 2023
> > > > [2] Masked Lip-Sync Prediction by Audio-Visual Contextual Exploitation in Transformers, SIGGRAPH Asia 2022
> > > > [3] DINet: Deformation Inpainting Network for Realistic Face Visually Dubbing on High Resolution Video, AAAI 2023
> > > > [4] SadTalker: Learning Realistic 3D Motion Coefficients for Stylized Audio-Driven Single Image Talking Face Animation, CVPR 2023
> > > > [5] StyleTalk: One-shot Talking Head Generation with Controllable Speaking Styles, AAAI 2023
> > > > [6] SPACE: Speech-driven Portrait Animation with Controllable Expression, Arxiv 2022
> > > > [7] Progressive Disentangled Representation Learning for Fine-Grained Controllable Talking Head Synthesis, CVPR 2023
> > > > [8] Pose-Controllable Talking Face Generation by Implicitly Modularized Audio-Visual Representation, CVPR 2021
> > > > [9] One-shot Talking Face Generation from Single-speaker Audio-Visual Correlation Learning, AAAI 2022

---

> > > > > ### Comment · Reviewer_BHq7 · 2023-08-21
> > > > >
> > > > > I would like to thank the authors for the extra metrics. I do not understand the decision to have Wav2Lip (i.e., a dubbing method) work with one frame and not use it in the way that they use their own method (which also does dubbing) and feel that this should have been done already for the first draft of the paper. In light of the rebuttal and responses from the authors I will upgrade my rating to borderline accept as I still feel there is no methodological novelty in the paper and the evaluation metrics are limited.
> > > > >
> > > > > There were several omissions in the original manuscript that misrepresented the method (e.g. No credit given to Make-a-Video and video-diffusion models despite using  the same temporal attention, not using the dubbing setting for Wav2Lip). If the paper is accepted I would recommend that these be revised in the final version of the manuscript.

---

> > > > > > ### Author Response · Authors · 2023-08-21
> > > > > >
> > > > > > Thank you for upgrading the rating. For Make-A-Video and Video Diffusion Models, which inspire us a lot when we design FaceComposer, we will add more discussions and the missing references. For Wav2Lip, we will follow and claim the dubbing setting in the final version.
> > > > > >
> > > > > > We really appreciate your efforts in the reviewing process, and we will revise our final version as you recommend.
> > > > > >
> > > > > > The Authors

---

### Official Review · Reviewer_BQjN · 2023-07-02

**Soundness:** 3 good
**Presentation:** 3 good
**Contribution:** 3 good
**Rating:** 6
**Confidence:** 4

**Summary:**

This paper presents a facial content generation framework named FaceComposer, which is based on a Latent Diffusion Model (LDM). The primary aim of this framework is to facilitate text-conditioned face synthesis/editing and animation. The conditions employed in this model encompass a variety of aspects, including mask, PNCC (Projected Normalized Coordinate Code), sketch, identity feature, and Text2Face embedding. To ensure the generation of dynamic content, the authors introduce a temporal self-attention module within the LDM during the training phase. The experimental results affirm the superiority of this framework, showcasing enhanced synthesis quality in both static and dynamic settings. This novel approach thus provides a robust solution to the challenges of face synthesis and animation.

**Strengths:**

1. The concept underlying this paper is both straightforward and efficacious, offering a user-friendly yet potent solution to the problems at hand.

2. The application of the Projected Normalized Coordinate Code (PNCC) as a condition in the diffusion model, particularly in the context of face animation, presents a unique and stimulating approach to this field of study.

3. The collection of a new dataset comprising more than 500 hours of talking face videos is an important contribution. This sizable dataset is likely to have significant utility in the further exploration and development of this area.

**Weaknesses:**

1. Comparative Methods: Including the results from StyleTalk [20] in the comparison would be beneficial given its shared design principle of audio2PNCC for talking face generation. This comparison could provide a more comprehensive overview of how the proposed method performs against closely related approaches.

2. Dataset Details: As Table 4 indicates, the release model of all comparison methods isn't trained on HDTF. This makes it difficult to ascertain whether the performance gap results from the FaceComposer itself or the additional training data sourced from Youtube, BBC, and so on. Conducting more ablation studies on the influence of the dataset would provide more clarity in this regard.

3. Missing References: It appears that there are some relevant references missing from the current list.

[a] DreamPose: Fashion Image-to-Video Synthesis via Stable Diffusion

[b] Pretraining is all you need for image-to-image translation

**Questions:**

1. As highlighted in the perceived weaknesses, the comparison settings for the different methods do not appear to be identical. To provide a fair and comprehensive evaluation, it would be beneficial to include some ablation studies to investigate the effect of dataset scale on generation quality. This additional layer of analysis would contribute significantly to the robustness and validity of the study's findings.

**Limitations:**

Certainly, the paper could be further enriched by discussing the social implications of the collected dataset. The acquisition of such an extensive dataset, with over 500 hours of talking face videos, has broad implications that could be relevant to many fields. It would be better to include some social impact discussion on it.

---

> ### Author Rebuttal · Authors · 2023-08-09
>
> Thank you for your comments!
>
> $\textbf{1. Comparative methods.}$
>
> Thanks. Table R1 displays the comparison results between FaceComposer and StyleTalk. It can be observed FaceComposer has slightly better performance than StyleTalk, stemming from two aspects: 1) FaceComposer employ FLAME to represent the geometric information of the face, which is more dense and expressive than the BFM used by StyleTalk [1]; 2) As the foundation model of FaceComposer, Latent Diffusion Models has greater potential in generation than the PIRenderer of StyleTalk.
>
> $\textbf{2. Dataset details and dataset scale.}$
>
> Firstly, considering FaceComposer may benefit from the same distribution between testing and training sets (both from HDTF) in Table 4, while other methods are not trained on HDTF, we add a test set MEAD-Neutral to remove the performance bias introduced in the testing stage, as shown in Table R1.
>
> Secondly, we agree with you that including some ablation studies of dataset scale will provide a comprehensive evaluation, so we verify the effect of dataset scale on face animation task in Table R4 (bottom half), other tasks are on the progress. We can observe that: 1) FaceComposer- is still better than other SOTA methods, proving the superiority of FaceComposer itself; 2) FaceComposer performs better than FaceComposer-, demonstrating that the large-scale dataset is essential for our FaceComposer, so we collect a large-scale high-quality training data to support different facial content creation tasks.
>
> $\textbf{3. Missing references.}$
>
> Thanks. We will add and discuss them in the final version.
>
> $\textbf{4. Impact of the collected dataset.}$
>
> Thanks. Our dataset will facilitate numerous research areas. The large data volume and high video quality of our dataset are well suited for various image/video generation tasks, such as unconditional face generation, face reenactment, and face swapping. Our dataset also has potential for 3D applications, such as 3D face generation. As a high-quality audio-visual dataset, our dataset facilitates the research in audio-visual speech recognition, speech separation, and audio-driven face animation.
> Our dataset can be used for deepfakes, which will have a negative effect. However, our dataset can also be leveraged in forgery detection tasks to prevent such concerns. We filter out harmful content from our dataset to prevent it from being used for malicious purposes. We will do our utmost to regulate the application and acquisition of our dataset to avoid potential misuse.
> We will include the discussion in the final version. Thanks.
>
> [1] Learning a model of facial shape and expression from 4D scans, SIGGRAPH Asia 2017

---

> > ### Comment · Area_Chair_8t4a · 2023-08-19
> >
> > Thanks to the authors for your response.
> >
> > @Reviewer BQjN: Does the rebuttal fully address your concerns?
> >
> > Best regards,
> > Your AC

---

### Official Review · Reviewer_tZ4V · 2023-07-04

**Soundness:** 3 good
**Presentation:** 3 good
**Contribution:** 3 good
**Rating:** 5
**Confidence:** 4

**Summary:**

The paper proposes a unified framework for facial generative models that allows text/spatial/audio condition facial editing tasks. The results are reasonable and comparable to prior works. Based on the stable diffusion prior, this model can generalize well to different style domains.

**Strengths:**

- The paper presents a unified framework that allows different face editing settings including face stylization, audio-driven animation, attribution editing, etc. The simultaneous style transfer and facial animation provides a one-stage solution to stylized facial animation, which avoids error accumulation and computation waste. More importantly, based on the SD prior, this model can generalize arbitrary style in the wild, such as anime, oil painting etc.

- The technical contribution is moderate as most of the modules come from prior works. However, the whole system aims at addressing several face generative tasks in one model which is beneficial for many applications.

- The paper organization and writing are easy to follow.

**Weaknesses:**

- The paper would benefit from an analysis of the pros and cons of jointly training with different conditions using text and experiment results. This is one key to support the effectiveness of the unified framework design.

- Some face generation+editing results in Figure A5 look odd. For example, the mask inpainted face has a different skin color compared to the neck in (a).

- While MakeItTalk, Wav2Lip and PC-AVS are not SOTA methods of face animation, recent related works such as styleTalk, styletalk, AVCT etc. should be discussed and compared if available.

**Questions:**

The paper would benefit from a discussion and comparison with recent related works. This would help address my concerns about missing comparisons and discussions with recent related works.

**Limitations:**

The social impact has been discussed.

---

> ### Author Rebuttal · Authors · 2023-08-09
>
> Thank you for your comments!
>
> $\textbf{1. Pros and cons of different conditions.}$
>
> It is agreed that the analysis of the pros and cons of jointly training with different conditions is important for our unified framework design, we list them below and will add them in the final version.
>
> $\textbf{Pros:}$ Training with multi-conditions, FaceComposer can support different tasks (e.g. face generation, face editing and face animation) and enjoy diverse controllabilities (e.g. accomplishing combined tasks among face creating, editing and animating with one-time forward) **with a unified model**. And Table R4 (upper half) shows that the performance of FaceComposer keeps stable when changing the number of conditions on a fixed dataset.
> $\textbf{Cons:}$ Due to the general design of FaceComposer for a variety of tasks, we need more training data to make different tasks perform well, which inevitably increases the cost of training. With a limited scale of training data, the performance of FaceComposer may degrade on some tasks, as illustrated in Table R4 (bottom half). We thus collect a high-quality large-scale multi-modal face database to alleviate such a problem.
>
> $\textbf{2. Problems in Figure A5.}$
>
> Thank you for pointing it out. In Figure A5(a), FaceComposer needs to fill the mask region in target face with the Identity Feature from source face. We admit this is a challenging setting since the Identity Feature only provides identity attribute, without facial skin or texture information. The skin color mismatch issue can be alleviated by masking both face and neck, and the visualization of this task could be affected by the difference between source and target face, such as age, gender. We show more results in Figure R2.
>
> $\textbf{3. More SOTA of face animation.}$
>
> Thanks for your reminder, we add more test sets, metrics and SOTA methods in Table R1. As we can see, FaceComposer achieves the best results, demonstrating the effectiveness of our design on face animation. StyleTalk is second to ours, because we share the same audio2PNCC design, but the representations of PNCC and generator of StyleTalk are inferior to FaceComposer. SadTalker adopts similar 3D motion coefficients regression, performing closely to StyleTalk. AVCT and EAMM are worse than the former three methods, since AVCT trains the model on one-identity dataset and the condition design of EAMM can not make full use of multi-identity dataset.

---

> > ### Comment · Area_Chair_8t4a · 2023-08-19
> >
> > Thanks to the authors for your response.
> >
> > @Reviewer tZ4V: Does the rebuttal fully address your concerns?
> >
> > Best regards,
> >
> > Your AC

---

> > ### Comment · Reviewer_tZ4V · 2023-08-20
> >
> > Thanks for the rebuttal. I have read other reviews and authors' feedback. The rebuttal has addressed all my concerns. More analysis and experiments have been presented. I would keep my initial rating.

---

> > > ### Author Response · Authors · 2023-08-21
> > >
> > > Thanks for your feedback and positive comments. We will improve our final version accordingly.
> > >
> > > The Authors

---

### Author Response · Authors · 2023-08-09
**Rebuttal Video Link**

We provide a rebuttal video link: https://drive.google.com/file/d/1BCMHcoTV7-Fi5FOHTypzcSsgBZU_upsp/view?usp=drive_link to demonstrate our generation results without mask condition.

---

### Author Rebuttal · Authors · 2023-08-09

To all reviewers:

We thank all reviewers for their efforts in reviewing our paper and appreciate their valuable comments. We will address their individual concerns in the rebuttals per review. Here, we list some concerns in common. If not specified, Table(Figure) \*/A\*/R\* represent the corresponding table(figure) in main paper/Supplementary Material/pdf file in the rebuttal, respectively.

$\textbf{1. More experiments of face animation in Table 4.}$

For face animation, more experimental comparisons are shown in Table R1. (1) We add four SOTA methods, including StyleTalk, AVCT[1], EAMM[2] and SadTalker[3]. It can be seen FaceComposer outperforms all of them. Note that despite sharing the design principle of audio2PNCC, FaceComposer is superior to StyleTalk, since the FLAME we use to represent PNCC is more expressive than the BFM that StyleTalk adopts [4], and the Latent Diffusion Models in FaceComposer has more potential in terms of generation than the PIRenderer in StyleTalk. (2) Considering the audio2PNCC of FaceComposer is trained on HDTF, we add another testset (MEAD-Neutral, a subset from MEAD with only neutral expression) to demonstrate our superiority across different data distributions. (3) Since some methods use mask (e.g. FaceComposer, Wav2Lip) and some use one-shot (e.g. PC-AVS, MakeItTalk, StyleTalk), in order to reach a fair comparison, we add two new metrics (SSIM-M and CPBD-M) besides M-LMD, to only evaluate the mouth region quality. SSIM-M and CPBD-M stand for SSIM and CPBD calculated in the mouth area, respectively.

$\textbf{2. Ablation study of condition numbers.}$

We add the ablation studies in Table R4 (upper half) to investigate the effect of different numbers of conditions on FaceComposer. Considering face generation/editing/animation are the basic tasks, we take FaceComposer with three conditions (T2F Embedding, Mask, PNCC) as baseline, "baseline + Sketch" means baseline with Sketch condition, "baseline + Sketch + Identity Feature" is equal to FaceComposer with all five conditions. It can be seen that Facecomposers with different number of conditions keep stable performance, no matter in face generation, editing or animation task. We argue this is reasonable, bacause the training dataset is fixed, when the number of conditions increases, no additional information is introduced for a specific task. Note that adding conditions does not increase the training set, but increases the number of conditions that are decomposed from the training set.

$\textbf{3. Ablation study of dataset scale.}$

To demonstrate the impact of dataset scale on generation quality, we show an ablation study on face animation in Table R4 (bottom half). More experiments on face generation/editing are currently being conducted and will be included in the final version. Considering the SOTA methods generally have dozens of hours of training data, we reduce the training data of FaceComposer to the similar scale for a fair comparison. Specifically, we randomly sampled 10 hours of video and 4.5W images from our original dataset to train a FaceComposer (denoted as FaceComposer-). From Table 4 (bottom half), it can be observed that FaceComposer- is inferior to FaceComposer due to the decrease of data information, but it is still better than other SOTA. Besides the performance gap caused by small dataset, we would like to clarify FaceComposer is a unified generative model that requires a large-scale training data to benefit a variety of facial content creation tasks. Therefore, we collect a large-scale multi-modal face database and will make it public under the constraints of ethical review.

[1] One-shot Talking Face Generation from Single-speaker Audio-Visual Correlation Learning, AAAI 2022
[2] EAMM: One-Shot Emotional Talking Face via Audio-Based Emotion-Aware Motion Model, SIGGRAPH 2022
[3] SadTalker: Learning Realistic 3D Motion Coefficients for Stylized Audio-Driven Single Image Talking Face Animation, CVPR 2023
[4] Learning a model of facial shape and expression from 4D scans, SIGGRAPH Asia 2017

---

### Decision · Program_Chairs · 2023-09-21

**Decision:**

Accept (poster)

**Comment:**

The paper received 2 Borderline Accept, 1 Weak Accept, and 1 Accept decision. The paper addresses several face-generative tasks in one model, which benefits many applications. The collection of a new dataset comprising more than 500 hours of talking face videos is also an important contribution. Although the original manuscript had several analyses missing, the authors presented them well in the rebuttal. Ethics Reviewers raise some concerns; the authors partially addressed them, and the paper can be released with proper care.

The ACs decided to accept the paper to NeurIPS. The authors should include the extra analyses presented in the rebuttal in the final paper version. The authors also should follow the recommendations from Ethics Reviewers.